# Universal radiation tolerant semiconductor

Alexander Azarov [1] ✉, Javier García Fernández [1], Junlei Zhao [2], Flyura Djurabekova [3], Huan He[3], Ru He[3], Øystein Prytz [1], Lasse Vines[1], Umutcan Bektas[4], Paul Chekhonin[4], Nico Klingner[4], Gregor Hlawacek [4] & Andrej Kuznetsov [1] ✉

Radiation tolerance is determined as the ability of crystalline materials to withstand the accumulation of the radiation induced disorder. Nevertheless, for sufficiently high fluences, in all by far known semiconductors it ends up with either very high disorder levels or amorphization. Here we show that gamma/beta (γ/β) double polymorph $Ga_2O_3$ structures exhibit remarkably high radiation tolerance. Specifically, for room temperature experiments, they tolerate a disorder equivalent to hundreds of displacements per atom, without severe degradations of crystallinity; in comparison with, e.g., Si amorphizable already with the lattice atoms displaced just once. We explain this behavior by an interesting combination of the Ga- and O- sublattice properties in γ-$Ga_2O_3$. In particular, O-sublattice exhibits a strong recrystallization trend to recover the face-centered-cubic stacking despite the stronger displacement of O atoms compared to Ga during the active periods of cascades. Notably, we also explained the origin of the β-to-γ $Ga_2O_3$ transformation, as a function of the increased disorder in β-$Ga_2O_3$ and studied the phenomena as a function of the chemical nature of the implanted atoms. As a result, we conclude that γ/β double polymorph $Ga_2O_3$ structures, in terms of their radiation tolerance properties, benchmark a class of universal radiation tolerant semiconductors.

Long-range periodicity or translation symmetry is a unique property of solids, even though solids may form amorphous phases too. In this context, accelerated particle beam irradiations are known to induce amorphization in many types of crystals, e.g. in semiconductors[1,2]. In its turn, radiation tolerance in semiconductors is determined as an ability to withstand the accumulation of the radiation disorder, otherwise leading to highly disordered lattice and, upon irradiating with sufficiently high fluences, to amorphization[3,4]. The irradiation-induced disordering mechanisms are generic, even though exhibiting material-specific differences, allowing us to classify semiconductors as low- or high-radiation tolerant[5–8]. Importantly, very recently, it was discovered that in gallium oxide ($Ga_2O_3$), which is a promising material for the next generation power electronics[9–13], the amorphization may

be prominently suppressed by the formation of a new metastable crystalline polymorph phase[14]. This process occurs in the irradiation interaction volume and results in a new polymorph film, separated from the initial polymorph by a sharp interface.

In this work, we report that such double polymorph $Ga_2O_3$ structures exhibit high radiation tolerance. Specifically, for room temperature experiments, these samples tolerated a disorder equivalent to hundreds of displacements per atom (dpa), without severe degradations of the crystallinity. For comparison, other semiconductors studied in literature for comparative dpa either amorphize or exhibit a high degree of lattice disorder. Notably, to induce such high dpa we use high fluence ion irradiations creating high excess of implanted atoms, maximized at the depth of the ion range, and

[1]University of Oslo, Centre for Materials Science and Nanotechnology, PO Box 1048 Blindern, N-0316 Oslo, Norway. [2]Department of Electrical and Electronic Engineering, Southern University of Science and Technology, Shenzhen 518055, China. [3]Department of Physics, University of Helsinki, P.O. Box 43, FI-00014 Helsinki, Finland. [4]Helmholtz-Zentrum Dresden-Rossendorf, D-01328 Dresden, Germany. ✉e-mail: alexander.azarov@smn.uio.no; andrej.kuznetsov@fys.uio.no

affecting the process depending on the chemical nature of the implanted atoms.

## Results and discussion

Figure 1 illustrates such high radiation tolerance of the double polymorph $Ga_2O_3$ structures, tolerating up to 265 dpa (see Supplementary Note 1 for the dpa calculations) without severe degradation in crystallinity (panels a–c), set in a context of the same characteristics in other semiconductors (panel d). Figure 1a plots the Rutherford backscattering spectrometry in channeling mode (RBS/C) spectra of the double polymorph $Ga_2O_3$ structures as a function of the fluence in the range of $1 \times 10^{16}$ to $1 \times 10^{17}$ Ni/$cm^2$. As explained in Methods and in more details in Supplementary Note 2, the characteristic shape observed for the $1 \times 10^{16}$ Ni/$cm^2$ RBS/C spectrum is a fingerprint of the high crystallinity of the double polymorph gamma/beta (γ/β) $Ga_2O_3$ structure. Thus, the data in the range of 300–400 channel numbers for $1 \times 10^{16}$ Ni/$cm^2$ implants correspond to the least disordered γ-$Ga_2O_3$ in Fig. 1a, which we adapt as a "reference" disorder level to compare with the data for higher fluences. Remarkably, the characteristic shape of the RBS/C spectra is maintained for $3 \times 10^{16}$ Ni/$cm^2$ and for $5 \times 10^{16}$ Ni/$cm^2$ implants, corresponding to 80 dpa and 132 dpa, respectively. Moreover, a minor deviation from the trend – related to

an enhanced RBS/C yield – observed for the $1 \times 10^{17}$ Ni/$cm^2$ implants (dpa = 265), is related to an increase in the Ni content, with no significant changes in crystallinity of the surrounding matrix. It is evident from the comparison of the corresponding channeling and random spectra in Fig. 1a and from the scanning transmission electron microscopy (STEM) data in Fig. 1b and c, that the sample retains exceptionally high crystallinity even after $1 \times 10^{17}$ Ni/$cm^2$ implants. In fact, the selected area electron diffraction (SAED) indexation of the sample (Fig. 1b) confirms its identification as γ-$Ga_2O_3$/β-$Ga_2O_3$ double-layer structure (see Supplementary Note 3). In its turn, we observed 3–6 nm diameter Ni precipitates embedded into the γ-$Ga_2O_3$ layer. These are shown in Fig. 1c with a magnified annular dark field (ADF)-STEM image taken along [100] γ-$Ga_2O_3$ axis, resolving the precipitates in a brighter contrast (see detailed analysis in Supplementary Note 4). Additionally, the γ/β $Ga_2O_3$ interface exhibits stacking with rather low lattice mismatch (Supplementary Note 5). Thus, based on the data in Fig. 1a–c, γ-$Ga_2O_3$ tolerates up to 265 dpa without severe degradations in the crystallinity of the semiconductor matrix exposed to Ni implants. This is a remarkable result in particular, when compared with literature data, see the data summarized in Fig. 1d. Indeed, as can be seen from Fig. 1d, such materials as Si, SiC, InP have previously been shown to amorphize already at 0.2–0.5 dpa[15–17], in contrast to the so-called high

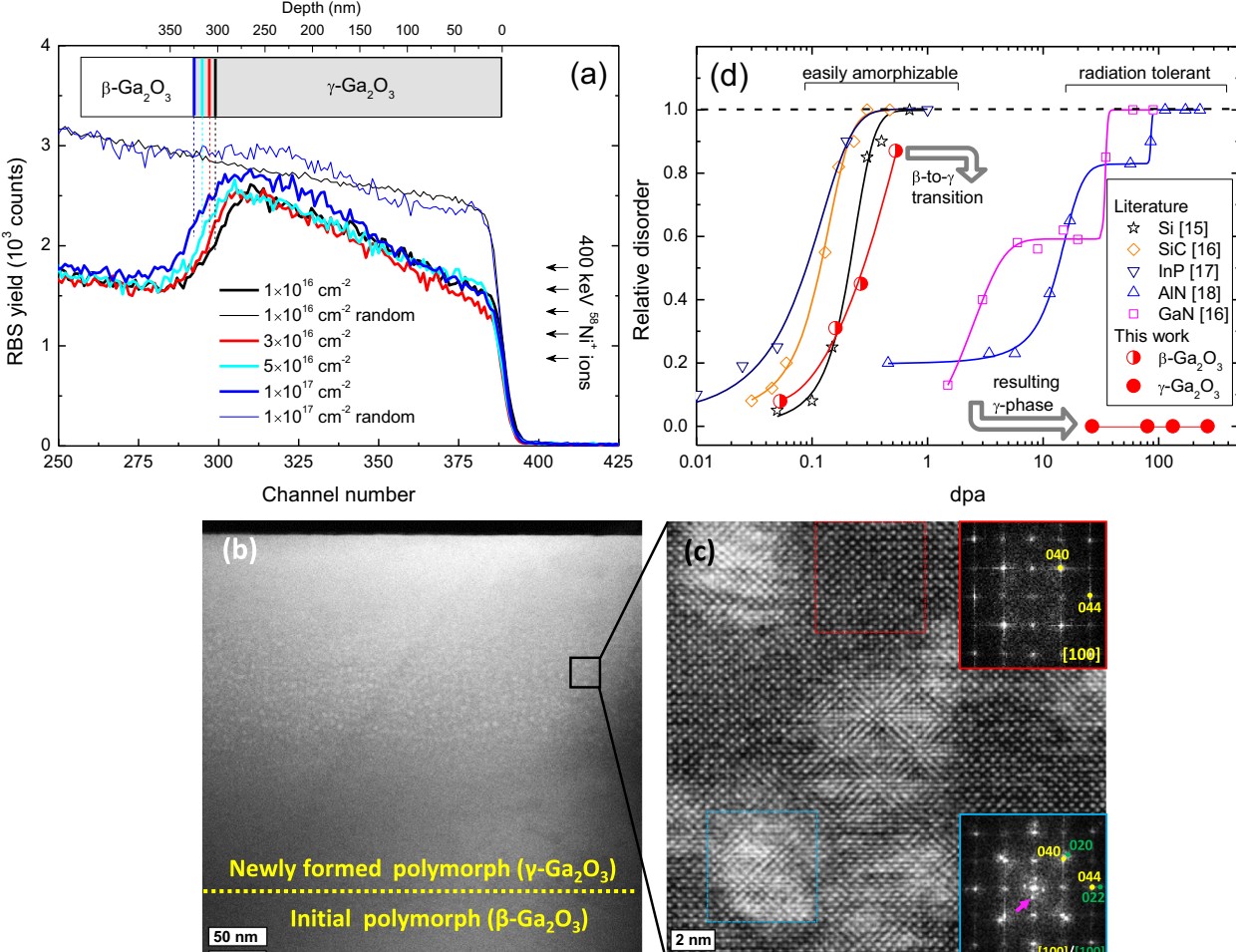

**Fig. 1 | High radiation tolerance of the double γ-$Ga_2O_3$/β-$Ga_2O_3$ polymorph structures. a** Random (thin lines) and channeling (thick lines) RBS spectra of (010) β-$Ga_2O_3$ samples implanted with 400 keV $^{58}Ni^+$ ions to the different fluences as indicated in the legend. **b** low magnification HAADF- STEM image of the $1 \times 10^{17}$ Ni/$cm^2$ sample showing full implanted region, (**c**) high resolution ADF-TEM and corresponding FFTs of the areas with (blue) and without (red) Ni precipitates. γ-$Ga_2O_3$ planes are indicated in yellow, metallic Ni in green and double diffraction spots are

indicated with a pink arrow; (**d**) relative disorder as a function of dpa for easily amorphizable (Si (stars)[15], SiC (diamonds)[16] and InP (down triangles)[17]) and radiation tolerant semiconductors (GaN (squares)[16], and AlN (up triangles)[18]) for Au implants at room temperature, as well as for $Ga_2O_3$ (this work, circles) - the lines are a guide to the eye. Notably, the depth scale in panel (**a**) is calculated for Ga atoms, so that the Ni related peak appears deeper in the sample (see Supplementary Note 2 for clarity). Source data are provided as a Source data file.

radiation tolerant materials, e.g. GaN or AlN, that are capable to accommodate much higher radiation disorder[16,18] and remaining crystalline. However, none of these materials remains such excellently crystalline as γ-Ga$_2$O$_3$, see Fig. 1d. Notably, β-Ga$_2$O$_3$ belongs to the low radiation tolerant group of materials. However, we observe that the disorder accumulation in β-Ga$_2$O$_3$ lattice does not result in full amorphization but triggers transformation to a new crystalline polymorph[14,19–21], as illustrated with arrows showing the trend for converting the irradiated β-Ga$_2$O$_3$ volume into radiation tolerant γ-Ga$_2$O$_3$/β-Ga$_2$O$_3$ double polymorph structure (see also Supplementary Note 2). Notably, there is a gradual increase in the γ-Ga$_2$O$_3$ thickness as a function of fluence, as highlighted in Fig. 1a by the corresponding dashed lines. This thickness increase is consistent with our hypothesis of the disorder induced β-to-γ-Ga$_2$O$_3$ transformations[14] and the data in Fig. 1a may be used to estimate the corresponding disorder thresholds (see Supplementary Note 6).

Further, the fact that the Ni content in Fig. 1b, c was sufficient for the precipitation, implies that one has to account for the chemical nature of the implanted atoms, potentially altering the defect accumulation and eventual amorphization processes in γ-Ga$_2$O$_3$, as it may occur in other materials too[8]. Thus, for comparison, we investigated these phenomena for several other ions, choosing elements having strongly different chemical capabilities to interact with the matrix atoms. For that matter, Fig. 2 shows examples of the STEM data taken upon the implants resulting in the same dpa range (86–88 dpa) for Au and Ga ions. Importantly, as seen from Fig. 2a, the same high radiation tolerance of the γ-Ga$_2$O$_3$/β-Ga$_2$O$_3$ double-layer structures is observed for the Au implants. The crystallinity of the new polymorph is confirmed by SAED patterns collected along the [100], [110], [111], [112] zone axes of γ-Ga$_2$O$_3$, shown in Fig. 2b–e, respectively (Supplementary Note 3). In contrast, for Ga ion implants we observed ~50 nm amorphous layer formed at the depth of 150–200 nm below the surface, see Fig. 2f. This region corresponds to the end of the range for Ga ions where the concentration of implanted Ga reaches the maximum.

Notably, Fig. 2g shows a high magnification ADF-STEM image of the interface between γ-Ga$_2$O$_3$ and the amorphous phase. The corresponding fast Fourier transforms (FFT)s in the insets of Fig. 2g confirm that γ-Ga$_2$O$_3$ is oriented along the [100] zone axis while the FFT in the amorphous phase shows the features that are characteristic of amorphous materials. Moreover, the difference in atomic coordination of γ-, β- and amorphous Ga$_2$O$_3$ phases is illustrated by the fine structure of the oxygen-K edge in the electron energy-loss spectroscopy (EELS) spectra shown in Fig. 2h. In particular, the oxygen K-edge is characterized by two peaks at 538 eV and 543 eV[22] and the relative intensity of these peaks apparently changes depending on the localization of the measurements[23], see Fig. 2h. Importantly, lowering the Ga fluence changes the situation back to excellently maintained crystallinity in the γ-Ga$_2$O$_3$/β-Ga$_2$O$_3$ double-layer structures. An additional cross-check with inert noble gas Ne implants confirmed the trends of this process (Supplementary Note 5). Altogether, the data in Figs. 1 and 2 (plus data in Supplementary Figs. 2–8) suggest that γ-Ga$_2$O$_3$ lattice indeed tolerates high values of dpa, in the order of hundreds, before it eventually breaks upon reaching even higher fluences accumulating very high concentration of implanted impurities.

Importantly, these experimental observations are in excellent agreement with the results of our theoretical modeling. The high radiation tolerance of γ-Ga$_2$O$_3$ is evident from the comparison of structural modifications caused by accumulated Ga-type Frenkel pairs (FPs) in β- and γ-Ga$_2$O$_3$ (Fig. 3). We present the results obtained by classical molecular dynamics (MD) simulations of thermal equilibration of the β- and γ-Ga$_2$O$_3$ lattices with increasing number of Ga-type FPs, using the recently developed machine-learned Ga-O interatomic potential[24]. We illustrate the ordering in phases by means of radial distribution functions (RDF), i.e., pair-wise radial distributions of atoms irrespective of atom species. RDFs of crystalline structures characteristically exhibit clear peaks at the pair distances corresponding to the coordination shells. RDFs of amorphous structures exhibit only short-range order (SRO) peaks since there is no long-range

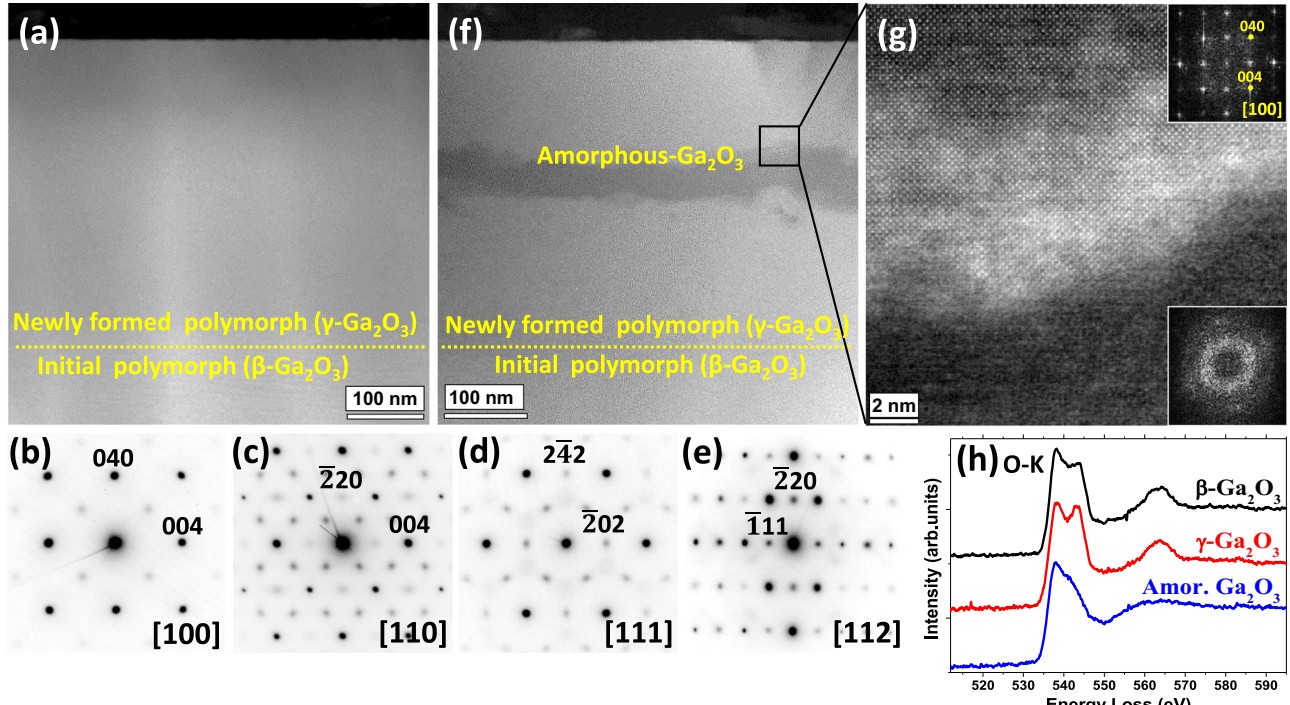

**Fig. 2 | Role of ion species in the radiation tolerance of γ-Ga$_2$O$_3$.** Low magnification HAADF-STEM images of the samples implanted with (**a**) Au and (**f**) Ga ions with the fluences corresponding to the 86–88 dpa range. SAED patterns of the γ-layer taken along [100], [110], [111] and [112] directions in the Au implanted sample are shown in the panels (**b**, **c**, **d**, and **e**) respectively. **g** High Resolution ADF-TEM image of the Ga implanted sample taken at the amorphous/crystalline interface with corresponding FFTs. **h** EELS spectra of the oxygen-K edge, acquired from β-, (black line) γ- (red line), and amorphous (blue line) Ga$_2$O$_3$ phases.

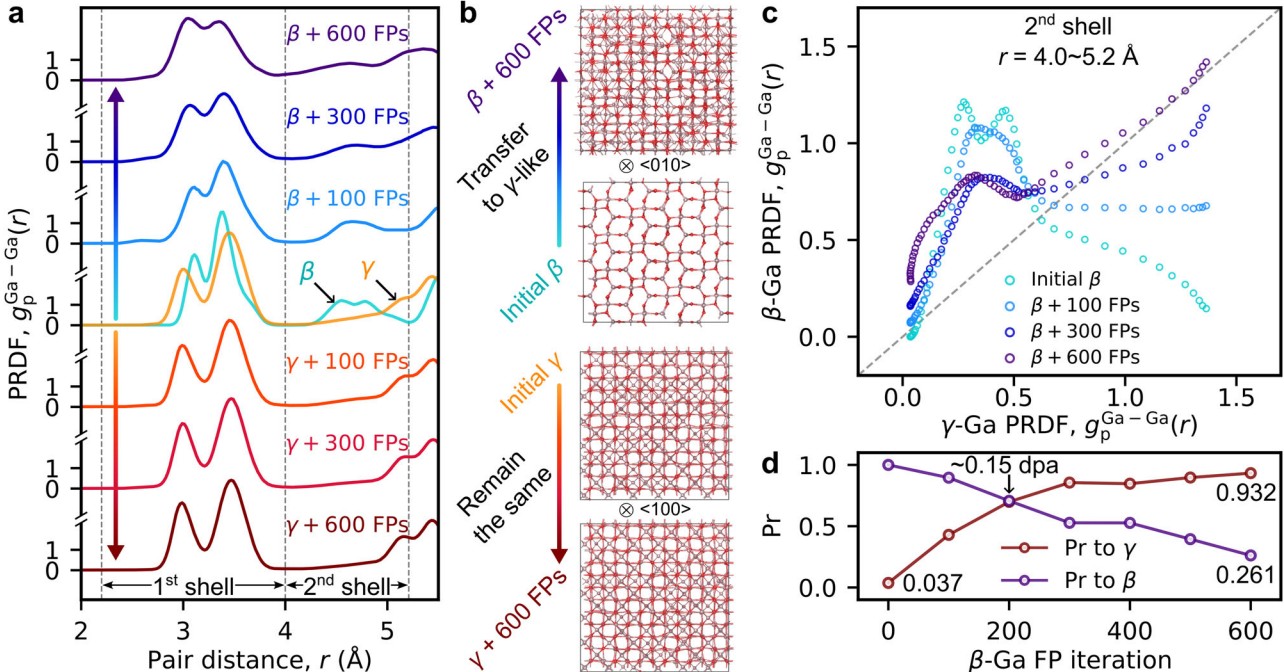

**Fig. 3 | Analysis of the PRDFs of Ga sublattices with additional Ga FPs in Ga₂O₃ lattices. a** Ga-Ga PRDFs for the pristine β- and γ-Ga₂O₃ lattices (in the middle); up and down from the pristine Ga-Ga PRDFs, the same PRDFs for the lattices with increasing numbers of FPs (up for β-Ga₂O₃ and down for γ-Ga₂O₃). For the analysis of structural modifications, the features of the Ga-Ga PRDFs are considered separately within the 1st (2.2 - 4.0 Å) and 2nd (4.0 - 5.2 Å) shells that are indicated by the vertical thin dashed lines. **b** The snapshots show modifications of both the β-Ga₂O₃ (up) and the γ-Ga₂O₃ (down) from the pristine lattices to the lattices with added 600 FPs. Ga ions are shown in brown and O in red. **c** The increasing similarity of the PRDF values of the β-Ga sublattice with increasing number of FPs versus the PRDF of the pristine γ-Ga within the 2nd shell. **d** The Pearson correlation coefficient, Pr, calculated within the 2nd shell for the PRDF of the increasingly damaged β-Ga with respect to the pristine β-Ga (blue circles) and γ-Ga (brown circles) PRDFs as a function of the FP number. The similarity to the γ-Ga phase is stronger above the threshold number of FPs (~200) compared to the similarity to the original β-Ga sublattice. See Supplementary Note 8, Supplementary Figs. 9–12 for more details. The link to the raw data is provided in the Data Availability Statement.

order (LRO) in these materials. For better insight, we plot the partial RDF (PRDF) within a specific atomic sublattice, i.e., the RDFs to the neighbors of a specific type (Ga-Ga, O-O or Ga-O) (see Methods). Notably, we focus on the evolution of the Ga-Ga PRDF in the β- and γ-Ga₂O₃ (β- and γ-Ga PRDFs, respectively), since the Ga sublattice responds to damage evidently the strongest compared to the O-O and Ga-O PRDFs, see Supplementary Note 8, Supplementary Fig. 8 for distinct differences in the Ga-Ga PRDFs and insignificant ones in the other two PRDFs with an increase in the number of FPs.

We show in Fig. 3a how the β- and γ-Ga PRDFs evolve with an increase of the number of FPs in the series of plots up (β-Ga) and down (γ-Ga) from the pristine β- and γ-Ga PRDFs shown together in the middle. The comparison reveals a prominent feature visible only within the 2nd shell in the β-Ga PRDF (peaks at ~4.5 Å), which are absent in the γ-Ga PRDF (Supplementary Note 8, Supplementary Fig. 10). Apparently, this feature vanishes and a shape characteristic to the γ-Ga PRDF becomes evident with an increasing number of FPs. The observed change manifests the β-to-γ Ga₂O₃ phase transformation with an increase of Ga-type defects in β-Ga₂O₃, while a similar damage level in γ-Ga₂O₃ does not result in any significant modification of the γ-Ga PRDF. Additionally, Fig. 3b illustrates the structural differences in β-Ga₂O₃ (up) and γ-Ga₂O₃ (down) before and after the introduction of 600 FPs, where the dramatic changes— compared to the initial cell— are seen only in β-Ga₂O₃ (for the more detailed transition process, see Supplementary Note 8, Supplementary Fig. 11). From quantitative comparison of the shapes of PRDFs within the 1st and the 2nd shells separately for both phases before and after introduction of Ga FPs (Supplementary Note 8, Supplementary Fig. 12), we deduce that only the damaged β-Ga PRDF within the 2nd shell underwent the most distinct shape modification. To compare these changes to the γ-Ga

PRDF, we map in Fig. 3c the β-Ga PRDF values for the different FP numbers against the corresponding values of the pristine γ-Ga PRDF. This analysis confirms that the shape of the damaged β-Ga PRDF with increase of Ga FPs indeed approaches that of the pristine γ-Ga PRDF. In Fig. 3d we plot the Pearson correlation coefficients (Pr) versus the numbers of FPs, comparing the shapes of the β-Ga at different number of FPs for the pristine β- and γ-Ga PRDFs (violet and brown Pr curves in Fig. 3d, respectively). The comparison reveals a high degree of positive correlation (similarity) for the damaged β-Ga PRDF with that of the γ-Ga PRDF after a threshold number of FPs, at ~200 FPs per cell (~0.15 dpa) when the β-Ga₂O₃ phase inevitably transforms into γ-Ga₂O₃ phase. This is in good agreement with the experimental data (Fig. 1d). Moreover, we see that the FPs have only marginal effect on the γ-Ga PRDFs, as shown in Fig. 3b and Supplementary Note 8, Supplementary Figs. 11–12, perfectly matching the strikingly high radiation tolerance of the γ-Ga₂O₃ observed in our experiments.

To verify the insensitive response of the O-O and Ga-O sublattice to the introduction of FPs observed in our MD simulations of damage accumulation, we performed dynamic single-cascade MD simulations, where the O and Ga atoms were naturally displaced in collision cascades. In these simulations, we see that the O sublattice is highly rigid and strongly prone to recrystallization into the face-centered-cubic (fcc) stacking, despite the stronger displacement of O atoms compared to Ga during the active periods of cascades, see Supplementary Note 8, Supplementary Fig. 13.

Furthermore, we study eventual chemical effects on the accumulation of structural disorder in γ-Ga₂O₃ using ab initio MD (AIMD). Figure 4a illustrates the PRDFs separately for the O-O and the heavy-ion sublattices compared to the respective initial PRDFs. The heavy-ion sublattice includes the native Ga and the added Ni, Au, or Ga atoms

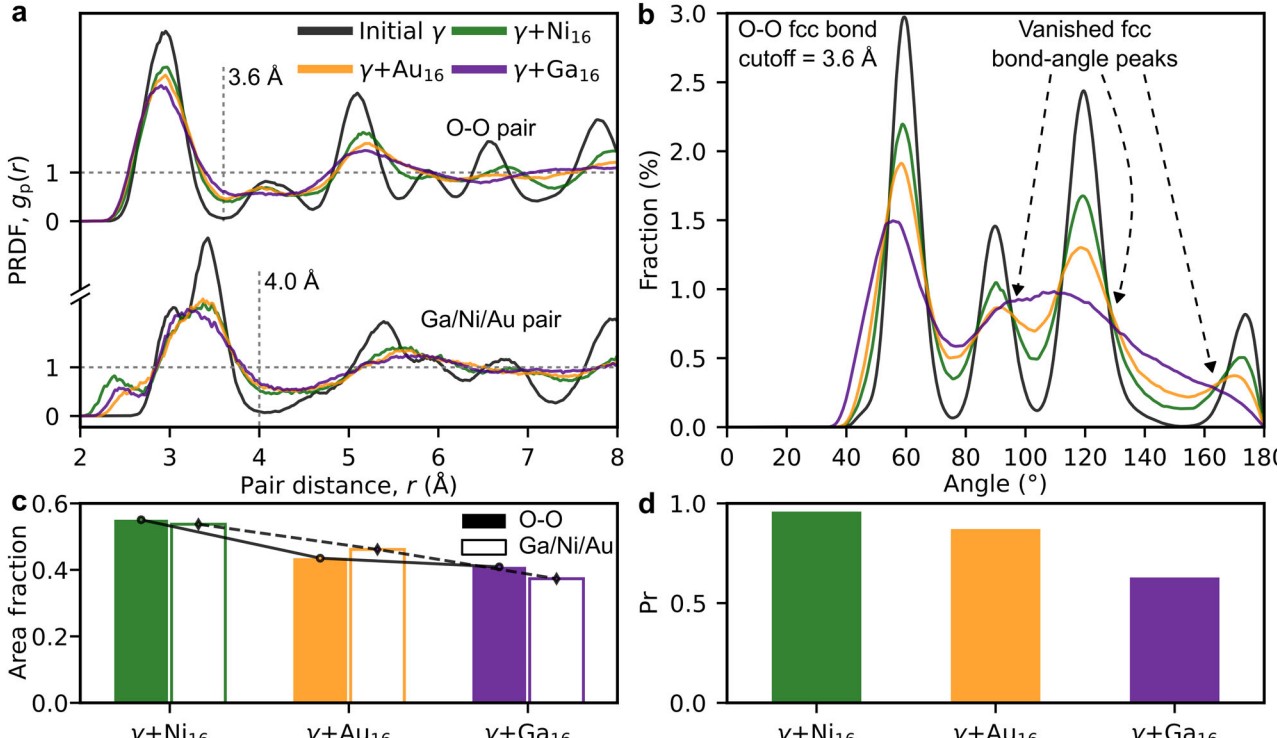

**Fig. 4 | Chemical effect of the foreign ions on the disordering of γ-Ga₂O₃ lattice.**
**a** AIMD-PRDFs of O-O and heavy-ion (Ga/Ni/Au) pairs at 900 K and 0 bar in the initial γ (black), γ + Ni (green), γ + Au (orange) and γ + Ga (purple). The first valleys are at 3.6 and 4.0 Å, as labelled by the vertical dashed lines. **b** Ratios of the absolute areas (covered by the PRDF curves with reference to 1): the distorted cells against the initial γ cell. **c** Bond angle distribution of O sublattice with O-O bond cutoff at 3.6 Å. **d** The Pr values of the distorted bond-angle distribution to the one of the initial γ cell. See Supplementary Note 8, Supplementary Figs. 14–15 for more details. The link to the raw data is provided in the Data Availability Statement.

(~10 at.%). In these simulations, the extra Ni, Ga, or Au atoms were added in random locations between the lattice sites imitating the implantation of ions under high-fluence irradiation. After that the structures were thermally equilibrated using AIMD to obtain the most energetically favorable structures (more data in Supplementary Note 8, Supplementary Figs. 14–15). Since the presence of LRO peaks in an RDF can be used as a crystallinity measure, we analyse both the O-O and heavy ion PRDF fluctuations around the unity (grey dotted line at g(r) = 1 in Fig. 4a) beyond the SRO peaks separated by the vertical grey dotted lines at distances 3.6 Å and 4.0 Å for the O-O and the heavy ions pairs, respectively. In Fig. 4b we quantify the degree of amorphization in the lattices with the implanted ions by integrating the total deviation area of the PRDF curves from the dotted lines (g(r) = 1) beyond the SRO peaks in Fig. 4a by the vertical grey dotted lines. The smaller the deviation area the higher the degree of amorphization the structure exhibits. Naturally, the PRDFs of the stoichiometric γ-Ga₂O₃ with multiple peaks and valleys along the g(r) = 1 line have the largest deviation area. Remarkably, the strongest disordering effect of the implanted atoms – the smallest deviation area – is observed in the cell with the Ga excess, which is in excellent agreement with the experiments (Fig. 2). However, the deviation area for the Ga-Au/Ga PRDF is only marginally larger than that of the Ga-Ga/Ga PRDF. Hence, we further analyse the disorder in the implanted lattices by comparing the bond-angle distributions for the O-O bonds for all three distorted structures with the pristine one in Fig. 4c and the Pr similarity analysis in Fig. 4d. The visual inspection of the plots in Fig. 4c reveals that the O-O bond angle peaks of the pristine lattice coincide with those of the Ni (green) and Au (orange) implanted structures, while the Ga-implanted cell does not exhibit similar O-O fcc bond-angle peaks beyond the first one at ~60°. Consistently, the Pr coefficients for the O-O bond-angle distributions in the Ni- and Au-implanted structures

are close to the unity, which shows a high degree of similarity with the corresponding distribution in the pristine γ-Ga₂O₃. In contrast, for the Ga-implanted structure Pr is only ~0.5, which essentially indicates the amorphization. This fact may be readily interconnected with the disturbance in the ionic charge distribution (charge transfer from the excess Ga atoms to the closest O ions) affecting the Coulombic interaction that maintains the order in an ionic system. Moreover, in Fig. 4a we see additional peaks in the green (Ni) and the purple (Ga) heavy-ion PRDF curves at 2.5 Å and 2.7 Å, respectively. These peaks can be correlated with metallic Ni precipitates observed in Fig. 1, while the Ga-Ga bonds may contribute to amorphization of the layer with the highest concentration of Ga ions in the Ga-implanted Ga₂O₃ in Fig. 2.

In conclusion, the full set of experimental and theoretical data in Figs. 1–4 may be seen as solid evidence for a discovery of the remarkable radiation tolerance in the γ-Ga₂O₃/β-Ga₂O₃ double-polymorph structures, practically independent of dpa. Meanwhile, the chemical effect introduced by high-fluence Ga ions leads to a nonstoichiometric disordered layer. This observation is rationalized by the unique combination of the specific features of both γ-Ga and O sublattices of γ-Ga₂O₃. Intrinsically defective, the γ-Ga sublattice is nearly insensitive to new point defects produced in collision cascades during ion irradiation, while the O sublattice is prone to rapid post-cascade recrystallization into original fcc stacking. The collaborative effect of both features explains macroscopically negligible structural deformations observed in heavily irradiated γ-Ga₂O₃.

## Methods
We used commercial (010) monoclinic beta Ga₂O₃ polymorph (β-Ga₂O₃) single crystals wafers from Tamura Corporation as initial polymorph substrates. To start with the samples were converted to double Ga₂O₃ polymorph structures with the implantation parameters

**Table 1 | Implantation parameters used in the present study**

| Ion | Energy (keV) | Fluence | | | $R_{pd}$ (nm) | $R_p$ (nm) | Max conc.(at.%) |
|---|---|---|---|---|---|---|---|
| | | (ions/cm²) | 1 dpa (ions/cm²) | (dpa) | | | |
| $^{58}Ni^+$ | 400 | $2 \times 10^{13}$–$1 \times 10^{17}$ | $3.8 \times 10^{14}$ | 0.05–265 | 115 | 160 | 0.001–5.8 |
| $^{197}Au^+$ | 1200 | $3 \times 10^{15}$, $1 \times 10^{16}$ | $1.2 \times 10^{14}$ | 26, 86 | 110 | 160 | 0.3, 0.9 |
| $^{69}Ga^+$ | 500 | $1 \times 10^{16}$, $3 \times 10^{16}$ | $3.5 \times 10^{14}$ | 29, 88 | 125 | 190 | 0.6, 1.9 |
| $^{20}Ne^+$ | 140 | $3.5 \times 10^{16}$ | $1.3 \times 10^{15}$ | 26 | 118 | 170 | 2.2 |

as reported in Ref. 14. For that matter we used $^{58}Ni^+$, $^{69}Ga^+$, $^{197}Au^+$, and $^{20}Ne^+$ ion implantation at room temperature, in particular adjusting implantation energies and fluences to obtain double polymorph $Ga_2O_3$ structures of comparative thickness while using different ions. Notably, all implants were performed at 7° off the normal direction of the wafer to minimize channeling. Furthermore, to avoid any heating of the samples during the implantations, the beam current was not exceeded 1 μA/cm² for Ni/Ne and 0.1 μA/cm² for Ga/Au implants. Table 1 summarizes the implantation parameters used in the experiments. Notably, the maximum of the nuclear energy loss profile ($R_{pd}$), the projected range ($R_p$), as well as the dpa values for each ion, were calculated using the SRIM code[25] simulations (see Supplementary Note 1). Table 1 also shows the ion fluences corresponding to 1 dpa in order to facilitate the fluence/dpa conversion for the readers. Importantly, upon each fluence collection step, the samples were measured by the RBS/C, while selected samples were also characterized with the STEM.

The RBS/C measurements were performed using 1.6 MeV $He^+$ ions incident along [010] $\beta$-$Ga_2O_3$ direction and 165° backscattering geometry. Importantly, it is known from the literature that upon the double polymorph $Ga_2O_3$ structure formation, the RBS/C yield exhibits a characteristic trend, attributed to the channeling conditions in the newly formed $\gamma$-$Ga_2O_3$ polymorph film - see Supplementary Note 2 for more details. This trend, if maintained as a function of the further fluence accumulation, is a fingerprint of the maintained crystallinity. Moreover, the horizontal scale in the RBS/C plots – the channel number – measures the thickness of the newly formed polymorph. Notably, Ga-parts of the RBS/C data were used in the analysis because of the significantly higher sensitivity of this method for heavier Ga-sublattice compared to the O-sublattice.

Further, STEM was used for detailed crystal structure and chemical analysis. For cross-sectional STEM studies, selected samples were thinned by mechanical polishing and by Ar ion milling in a Gatan PIPS II (Model 695), followed by plasma cleaning (Fishione Model 1020) immediately before loading the samples into a microscope. High Resolution Scanning Transmission Microscopy (HRSTEM) imaging, SAED, energy dispersive x-ray spectroscopy (EDS), and EELS measurements were done at 300 kV in a Cs-corrected Thermo Fisher Scientific Titan G2 60–300 kV microscope, equipped with a Gatan GIF Quantum 965 spectrometer and Super-X EDS detectors. The STEM images were recorded using a probe convergence semi-angle of 23 mrad, a nominal camera length of 60 mm using three different detectors: high-angle annular dark field (HAADF) (collection angles 100–200 mrad), annular dark field (ADF) (collection angles 22–100 mrad) and bright field (BF) (collection angles 0–22 mrad). The structural model of both phases was displayed using VESTA software[26].

EBSD was performed on the Ne irradiated sample in a Zeiss NVision 40 scanning electron microscope (SEM) equipped with a field emission electron cathode and a Bruker EBSD system with an e- Flash HR+ detector. To ensure the removal of a possible carbon contamination layer, the sample was cleaned for 45 s in an air plasma cleaner. The acceleration voltage was set to 30 kV, the beam current to about 10 nA using a 120 μm aperture. In order to record low noise high quality EBSD patterns, the detector resolution was set to 800 × 570 pixels and the exposure time to 8 × 122 ms per frame. EBSD was done as mappings of 20 × 15 steps, with a step size of 1.9 μm on the irradiated and the unirradiated surface sections of the sample

The ab initio molecular dynamics (AIMD) simulations were conducted using the Vienna Ab-initio Simulation Package (VASP)[27], employing the projected augmented-wave method[28]. The Perdew-Burke-Ernzerhof version of the generalized gradient approximation was used as an exchange-correlation functional[29]. The electronic states were expended in plane-wave basis sets with an energy cutoff of 400 eV throughout all AIMD runs. The Brillouin zones were sampled with a single-$\Gamma$ k-point for a $1 \times 2 \times 4$ 160-atom $\beta$-$Ga_2O_3$ supercell, and a $\Gamma$-centered $2 \times 2 \times 1$ k-mesh for a $1 \times 1 \times 3$ 160-atom $\gamma$-$Ga_2O_3$ supercell. In these simulations, the increase of experimental fluence was mimicked by introducing implanted atoms (Ni, Au, or Ga) in interstitial and substitutional (specified by the superscript $S$) lattice sites. Specifically, we added 8, 12, and 16 atoms of a given species, which corresponded to 5 at.%, 7.5 at.%, and 10 at.% concentrations with respect to the initial number of atoms in the cell. Initially, the obtained structures were relaxed to the local energy minimum with and without constraining the volume of the cell. Then, the relaxed cells were used in AIMD simulations to enable the dynamic evolution of the system to accommodate the added atoms in the best possible configurations. These simulations were performed for 5 ps with the step of 2 fs in isothermal-isobaric ensemble[30] at 900 K and 1 bar, employing Langevin thermo- and barostats[31].

The large-scale classical MD simulations were conducted using LAMMPS package[32]. The newly developed machine-learning interatomic potential of $Ga_2O_3$ system was employed[24]. The potential is developed to guarantee the high accuracy for $\beta$/$\kappa$/$\alpha$/$\delta$/$\gamma$ polymorphs and universal generality for disordered structures. In these simulations, Ga FPs were generated cumulatively in 1280-atom $\beta$-$Ga_2O_3$ and a 1440-atom $\gamma$-$Ga_2O_3$ cells by iteratively displacing a random Ga atom following a randomly directed vector with the norm of 10 - 15 Å. The two systems then firstly were relaxed to the local minimum to avoid initial atom overlapping and secondly were thermalized with NPT-MD for 5 ps at 300 K and 0 bar. In total, 600 Ga-FP iterations were run for both cells. In addition, we have performed MD simulations of single cascades in $\beta$-$Ga_2O_3$ at 300 K. The initial momentum direction and the position of a primary knock-on atom (PKA) were selected randomly at the center of the simulation cell. The PKA was assigned the kinetic energy of 1.5 keV. Periodic boundary conditions were applied in all directions. The temperature was controlled using a Nosé-Hoover thermostat[33] only at the borders of the simulation cell to imitate the heat dissipation in bulk materials. To avoid the cascade overlap with temperature-controlled borders, the number of atoms in the simulation cell was increased to 160 000. We applied the adaptive time step[34] for the efficiency of MD simulations in the active cascade phase. Electronic stopping as a friction term was applied to the atoms with kinetic energies above 10 eV. The simulation time of the single cascades was 50 ps. 120 simulations with different PKA were carried out for statistical analysis.

The structural modifications due to accumulated damage in the studied lattices were analyzed using radial distribution functions.

The RDF is defined as the ratio of the ensemble-average local number density of particles, $\langle\rho(r)\rangle$, at a distance $r$ from a reference particle to the average number density of particles in the system.

$$g(r) = \frac{\langle\rho(r)\rangle}{N_{at.}/V}, \tag{1}$$

where $N_{at.}$ is the total number of particles, and $V$ is the system cell volume. Essentially, the RDF is a fingerprint descriptor of the structural property of a system of particles down the atomic scale. For a crystal structure, this function is characterized by well pronounced peaks at the radial distances corresponding to the radii of coordination shells. While the SRO peaks are practically always present in a structure, the LRO peaks in amorphous structures are indistinguishable, since the number density of the atoms in the spherical shells at long distances in an amorphous structure is the same as the average number density in the structure. This feature of RDF gives a good measure of crystallinity in the studied structures.

The PRDF describes the type-specified sublattice in a multi-species system[35]. We performed a detailed analysis of the Ga-Ga, Ga-O, and O-O PRDFs as a function of stochastically generated FPs. All RDF distributions were obtained by averaging the signals between 2 to 5 ps for the frames recorded at every simulation step. For clarity, we discriminated the Ga-Ga PRDF features within so-called 1st and 2nd shells. The division was based on the significance of changes that are observed in PRDF distributions before and after the damage accumulation. The border was selected at the valley at ~4.0 Å. The Pearson correlation coefficient, Pr, between any two given curves is calculated with the formula:

$$\text{Pr}\,\frac{\sum_{i=1}^{n}\left(A_i - \bar{A}\right)\left(B_i - \bar{B}\right)}{\sqrt{\left[\sum_{i=1}^{n}\left(A_i - \bar{A}\right)^2\right]\left[\sum_{i=1}^{n}\left(B_i - \bar{B}\right)^2\right]}}, \tag{2}$$

where $A_i$ and $B_i$ are the variable samples of the two curves, respectively, and $\bar{A}$ and $\bar{B}$ are the mean values of the variable samples, respectively.

## Data availability

The data that support the findings of this study are available within the paper (and its Supplementary Information file) and from the corresponding authors upon request. Source data are provided with this paper. The machine-learning potential parameter files used to run classical MD simulations are openly available at https://doi.org/10.6084/m9.figshare.21731426.v1. The corresponding raw data of the classical and ab initio MD published in this paper are openly available at https://doi.org/10.6084/m9.figshare.23599950. Source data are provided with this paper.

## Code availability

The code and software used in this work are LAMMPS, VASP, OVITO, and SRIM which are openly available online from the corresponding developers and maintainers.

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

## Acknowledgements
M-ERA.NET Program is acknowledged for financial support via GOFIB project (administered by the Research Council of Norway project number 337627 in Norway, the Academy of Finland project number 352518 in Finland, and the tax funds on the basis of the budget passed by the Saxon state parliament in Germany). The experimental infrastructures were provided at the Norwegian Micro- and Nano-Fabrication Facility, NorFab, supported by the Research Council of Norway project number 295864, at the Norwegian Centre for Transmission Electron Microscopy, NORTEM, supported by the Research Council of Norway project number 197405. This work was also partly financed by the UiO Growth House funds in Norway. Support from the Ion Beam Center at the HZDR is acknowledged too. Computing resources were provided by the Finnish IT Center for Science (CSC) and by the Center for Computational Science and Engineering at the Southern University of Science and Technology. The computational work was also partially supported by Guangdong Basic and Applied Basic Research Foundation under Grant 2023A1515012048. The international collaboration was also fertilized via the COST Action FIT4NANO CA19140, supported by COST (European Cooperation in Science and Technology, https://www.cost.eu/), and via INTPART Program at the Research Council of Norway project number 322382.

## Author contributions
A.K. and A.A. conceived the research strategy and designed the methodological complementarities. A.A., J.G.F., U.B., P.C., and N.K. carried out experiments and provided initial drafts for the description of the experimental data. H.H., R.H., and J.Z. performed molecular dynamics simulations. F.D. and J.Z. developed the theoretical models and composed the theoretical part of the manuscript. A.K. and A.A. finalized the manuscript with input from all the co-authors. All co-authors discussed the results as well as reviewed and approved the manuscript. A.K., L.V., Ø.P., F.D., and G.H. administrated their parts of the project and contributed to the funding acquisition. A.K. coordinated the work of the partners.

## Competing interests
The authors declare no competing interests.
