## [Peer Review File · Nature Communications]

Universal radiation tolerant semiconductorREVIEWER COMMENTS

Reviewer #1 (Remarks to the Author):

The paper "Universal radiation tolerant semiconductor" reports an interesting experimental and simulation study of ion implantation in beta-Ga₂O₃. The authors report a phase transition from the beta phase to the gamma phase for high fluence implantation. The gamma phase is shown to be very radiation resistance showing no amorphisation (unless triggered by chemical effects) or other phase transition for considerably high implantation fluences. In particular, the good agreement between experiment and Molecular Dynamics simulations is remarkable. The results are interesting from fundamental point of view but also considering the great potential of the emerging Ga₂O₃ semiconductor for device applications. I therefore believe this paper should be published after considering the following comments:

1. The title "Universal radiation tolerant semiconductor" is quite pompous and does not really reflect the findings of the paper. There are other semiconductors with similarly high radiation tolerance (e.g. ZnO does not suffer amorphisation for similar ion fluences as used in the present study, even below room temperature). Furthermore, the title implicitly suggests that this semiconductor would be a universal solution for radiation resistant devices, but only structural aspects are discussed and no electrical, optical, magnetic properties which would enable device applications. As mentioned before, the strong point of the paper is the excellent agreement between experiment and simulation for a quite complex system including phase transitions and chemical effects. This should be reflected in the title. If gamma Ga₂O₃ is an interesting polymorph for device applications remains to be shown.
2. The sample implanted to a fluence of 1×10^{16} Ni/cm² that triggered the phase transition is repeatedly called "zero disorder" or "virgin gamma-Ga₂O₃" both in the manuscript and in the supplement. I think these names are misleading since they are suggesting that the gamma-phase has an almost perfect crystal quality. This is not reflected by the high minimum yields found in the RBS/C measurements. In the supplement, the authors attribute the high minimum yield to the low-index channel measured normal to the sample surface. This should be proven e.g. by doing RBS/C analysis along the inclined b- or c-axes which according to Fig. S5 should be easily found at a 45° inclination angle. TEM analysis is often not sensitive to some defect types, in particular point defects, so the high crystalline quality should be proven by some macroscopic technique such as RBS/C or X-ray diffraction. The term "zero disorder" or "virgin" could be replaced by reference spectrum.
3. Related with the previous point, it is important to make clear that "radiation resistance" in the present paper is used in the sense of high threshold for amorphisation. Probably the authors could comment on displacement energies for different phases based on MD simulations. Also, it would be interesting to review literature on some ceramics (nuclear materials), where high amorphisation thresholds are found for spinels and irradiation of pyrochlores can lead to amorphisation or phase transition depending on the cation and anion radii.
4. To the best of my knowledge, the first mentioning of a possible phase transition of (010) beta Ga₂O₃ upon ion implantation was published by Wendler et al. (NIM B 379 (2016) 85–90) and should be cited here. On the other hand, Lorenz et al. (Proc. SPIE 8987 (2014) 89870M) reported quite distinct RBS/C pattern in (100) oriented beta Ga₂O₃, where the aligned spectra reach the random level suggesting amorphisation starting at the surface. The authors themselves have published previous papers showing distinct strain accumulation upon ion implantation in Ga₂O₃ with different surface orientation. How can these results be related with the mechanism reported in the present paper?
5. It would help to more easily understand the paper if the crystallographic relationship between beta and gamma phases would be mentioned explicitly early in the manuscript (e.g. on page 6 when discussing SAED patterns). Also, the orientation of the original crystal should be mentioned earlier (at this moment it is only mentioned in the methods section).
6. I find Fig. 3c quite confusing to read and you are losing the information on pair distance in this representation. In fact, from this figure it seems that similarities are only strong in the end of the 2nd shell window. Would it be clearer to plot the ratio PRDF_beta/PRDF_gamma as a function of the pair distance? It should be close to 1 after phase transformation. Similar figures are also shown in the supplement.
7. The two texts (main manuscript and supplement) would be easier to read with less cross references (which force the reader to read both documents in parallel). E.g. in Fig. 1S one could

add the implantation energies in the figure labels (or caption) or in Fig. S7 one could repeat the color code in the caption instead of referring to the main manuscript.

8. Supplement page 10: "The fractional numbers indicate that there is an artefact related to the misalignment..." Not clear what the authors want to say by "artefact".

9. Supplement page 10: "(i) interface stacking in γ/β -Ga₂O₃ double polymorph structures is independent on the ion type used in fabrication process" Not clear from which technique you draw this conclusion since EBSD patterns are only shown for one sample.

10. Supplement page 11: "Nevertheless, there should be a reason for the DPA threshold overestimation deeper in the bulk in Fig. S5." Not clear what you want to say here. I don't think you want to say that SRIM overestimates DPA in depth.

11. It seems that Fig. S11 is not mentioned in the text.

12. Supplement Fig. S13: "to exclude coloring of atoms around single vacancies, which do not distort the lattice" Maybe I do not understand what authors want to say here, but there can be a strong relaxation of the lattice around a vacancy.

13. Some typos should be corrected: e.g. interface instead of interphase; fluences instead of fluencies; page S17: caused by the cumulated (instead of causing); page S18 Fig. S13 shows the MSD (not Fig. S12); page S19: in the same way as that in Fig (instead of than of that); page 10 Coulombic (instead of Columbic); page 15 density of particles in the system

Reviewer #2 (Remarks to the Author):

The authors report a very significant difference in radiation hardness, as quantified by resistance to accumulation of lattice disorder, in the little-studied gamma polytype of Ga₂O₃ compared to the most stable beta polytype. They do this in a systematic fashion, by varying the level of displacements per atom using different implanted ions. Once formed by a transition from the initial beta polymorph, the resultant gamma polymorph shows a very high level of resistance to further transformation. It's an important basic science result, since the level of radiation hardness exceeds that of AlN, which is very difficult to amorphize under normal conditions. From a practical viewpoint, it is less clear as to the significance, because it is known that the gamma polytype can be thermally transformed back to the beta phase.

A few comments that can add to the clarity of the manuscript.

1. they use values for the displacement energies of Ga and O that are relatively close to each other, but a recent paper using molecular dynamics simulations, based on first principles density-functional methods, to determine the nature and stability of the defects generated by atoms knocked-out by particle irradiation at near threshold energies (found to be 28 ± 1 eV for Ga and 14 ± 1 eV for O). Blair R. Tuttle, Nathaniel J. Karom, Andrew O'Hara, Ronald D. Schrimpf, and Sokrates T. Pantelides

, "Atomic-displacement threshold energies and defect generation in irradiated β -Ga₂O₃: A first-principles investigation", Journal of Applied Physics 133, 015703 (2023).

<https://doi.org/10.1063/5.0124285>

2 the damage accumulation in WBG and UWBG semiconductors is known to be very temperature-sensitive. You need to discuss more details on how the temperature of the samples was controlled during the implantation steps. Given the low thermal conductivity of the beta Ga₂O₃, if you are seeing significant temperature excursions, it could imply the gamma polytype is above its threshold temperature where amorphization cannot occur, as also happens with GaN and AlN.

3. a comment might be added about the Ni precipitates formed. Just in general, the very high doses used are above any of those typical for device fabrication, which means that the high radiation resistance gamma region would not have much utility, since it is both non-stoichiometric and thermally unstable.

4. SRIM has limitations in that it doesn't treat defect recombination during dynamic annealing. Please add a brief comment about the expected discrepancy in dpa from SRIM versus what might actually be the accumulated disorder level. Can you use your RBS spectra to make this clearer?

5. What is the role if any in electronic excitation versus nuclear displacements, since most of the energy deposited is from the former.

Overall, this is a very well-written manuscript and details a novel result. I recommend it's publication after the clarifications noted above.

Reviewer #3 (Remarks to the Author):

This is a well-written and technically sound manuscript. The discovery of the high radiation resistance of the gamma-/beta double polymorph of Ga₂O₃ is a highly noteworthy result and has not been previously reported. This double polymorph of Ga₂O₃ is probably the most radiation resistant wide bandgap semiconductor reported to date. This could have significant impact in the area of opto-electronic devices for space, nuclear and defense applications. Also noteworthy is the explanation provided for the transformation of beta Ga₂O₃ single crystal wafers to the double polymorph. This understanding could potentially lead thin film synthesis experts to directly synthesize the double polymorph of Ga₂O₃, although the ion-implantation approach is standard in the semiconductor industry and thus industrially feasible. The conclusions are well supported by the analysis of the experimental data and computational simulations that are provided in more detail in the supplementary material. The experimental and computational methodologies are sound and provide sufficient detail for the work to be reproduced, as well as confidence in the conclusions reached.

One technical comment related to the sentence on page 8: "In these simulations, we see the O sublattice is highly rigid and strongly prone to recrystallization into face-centered-cubic (fcc) stacking, despite higher mobility of O atoms compared to Ga during the active phase of cascades". Is it "despite the higher mobility of O atoms" or is it because of the higher mobility of the O atoms? The latter may be the more correct explanation. This is a little confusing, and the authors should consider rewording to provide clarity.

A minor comment is that "displacements per atom" is a unit of damage dose and generally written in lower case "dpa" not upper case "DPA". In Figure S1(a), the equation for DPA should be written as a function of Fluence, not Dose. Likewise, in the inset in Figure S1(b), the dpa should be a function of Fluence (cm⁻²), not Dose. This should also be corrected in the associated Supplementary Material text. The authors use the term "fluence" elsewhere and may confuse readers by substituting the term "dose" for "fluence". They are not the same.

Some minor editorial suggestions are provided in the attached file.

RESPONSE TO REVIEWERS' COMMENTS

Response on the comments of the Reviewer #1:

Comment: 1): *“The title “Universal radiation tolerant semiconductor” is quite pompous and does not really reflect the findings of the paper. There are other semiconductors with similarly high radiation tolerance (e.g. ZnO does not suffer amorphisation for similar ion fluences as used in the present study, even below room temperature). Furthermore, the title implicitly suggests that this semiconductor would be a universal solution for radiation resistant devices, but only structural aspects are discussed and no electrical, optical, magnetic properties which would enable device applications. As mentioned before, the strong point of the paper is the excellent agreement between experiment and simulation for a quite complex system including phase transitions and chemical effects. This should be reflected in the title. If gamma Ga₂O₃ is an interesting polymorph for device applications remains to be shown.”*

Reply: We have carefully considered the criticism regarding the title. However, even though we understood the arguments, we cannot agree with the reviewer statement on that the title is “pompous and does not really reflect the findings of the paper”.

Firstly, the title is not “pompous”. The definition of “pompous” in a dictionary is “arrogant or egotistic” (as defined with the closest synonyms, e.g. at the thesaurus.com). There are no aspects of that kind in the title “Universal radiation tolerant semiconductor”. On contrary, we think this title is clear and good for attracting attention of not only experts working in the specific field of this paper, but also for raising interest from a broader scope audience.

Secondly, in our opinion, it is exactly summing up the findings of the paper in a short way. Indeed, universal means “entire” or “broad” (e.g. as defined in thesaurus.com again by the closest synonyms), however with a superlative accent of course. This accent (belonging to wording “universal”) is appropriate since the level of the radiation tolerance we report in this paper is exceptional (even if comparing with other radiation tolerant materials as pointed out by the reviewer in the criticism and as has been taken into account in our paper by comparing different materials in Fig.1(d)). Indeed, the discovery of this paper is not in finding a new system comparable to e.g. ZnO which “does not suffer amorphisation for similar ion fluences”, but still displaying the accumulation of a very high degree of disorder. On contrary, we report a system that remains highly crystalline upon very high number of primarily displacements created in the lattice, leaving it “entirely” or “universally” crystalline. These arguments explain the use of the wording “universal” in the title.

Further, in our paper from the very beginning we clearly define that the “radiation tolerance” is considered in the structural terms, with no other explicit or implicit options. We agree with the reviewer, it would make a great added value to demonstrate radiation stability of the physical properties of this system for its use in, e.g. electronics. We are working in this direction and, by sharing the present data related to this unique structural radiation stability, we open this research avenue for other groups to join the efforts. In this sense, we fully agree with the statement that whether “ γ -Ga₂O₃ is an interesting polymorph for device applications” remains to be shown; however, we cannot see why it might prevent to classify this material as “radiation tolerant” if we know it is - in terms of its structural properties. Additionally, it must be noted that there are research fields beyond electronics, e.g related to nuclear applications, where the radiation tolerance is related to the modification of mechanical properties, primarily interconnected to the ability of materials remaining crystalline under irradiation.

At this end, we appreciate the reviewer pointed out towards that “the strong point of the paper is the excellent agreement between experiment and simulation for a quite complex system including phase transitions and chemical effects,” We agree with that formulation. However, we think, it is more appropriate for this broad scope publication to keep the original title – which still fully explains the prime observation made in the paper – instead of designing a new title with more details included.

Comment: 2): *The sample implanted to a fluence of $1e16$ Ni/cm² that triggered the phase transition is repeatedly called “zero disorder” or “virgin gamma-Ga₂O₃” both in the manuscript and in the supplement. I think these names are misleading since they are suggesting that the gamma-phase has an almost perfect crystal quality. This is not reflected by the high minimum yields found in the RBS/C measurements. In the supplement, the authors attribute the high minimum yield to the low-index channel measured normal to the sample surface. This should be proven e.g. by doing RBS/C analysis along the inclined b- or c-axes which according to Fig. S5 should be easily found at a 45° inclination angle. TEM analysis is often not sensitive to some defect types, in particular point defects, so the high crystalline quality should be proven by some macroscopic technique such as RBS/C or X-ray diffraction. The term “zero disorder” or “virgin” could be replaced by reference spectrum.*

Reply: Upon consideration, we decided to follow the reviewer recommendation to remove the wordings “zero disorder” and “virgin γ -Ga₂O₃” from the manuscript and specify those as “reference” states/spectra, to avoid further discussions with the reviewer on whether “ γ -phase has an almost perfect crystal quality” or not (hopefully avoiding further delays with the publication of this important paper). Otherwise, the reviewer understood it perfectly correct that the “high minimum yields found in the RBS/C measurements” for the γ -phase is because of “the low-index channel measured normal to the sample surface”. At the infrastructure we have in-house, we cannot rotate the sample up to the angles with better channels, we tried it. We have applied for the beamtime at a large-scale infrastructure allowing to measure at higher angles. If to wait until these data are collected and added to the present manuscript – it will imply significant delays with the publication of this paper, which would be unfortunate and, we think, unnecessary. We suggest the reviewer to consider our argumentation below. In our opinion, even though the comparison of the RBS/C data collected in γ -phase along different channels is spectacular, it will not add or change to the main message of this discovery/paper. Importantly, even though the RBS/C was a prime tool for systematic analysis of many tens of samples used altogether in this study, the ultimate prove of the high radiation tolerance comes from the electron microscopy studies – indeed showing very high crystalline quality of the γ -phase. This also was confirmed by the epitaxial relationship between β/γ phases from both TEM and EBSD results as clearly shown in Suppl. Mat V. Additionally, the capabilities of the modern STEM analysis might not be underestimated. Indeed, as the reviewer pointing out “TEM analysis is often not sensitive to some defect types, in particular point defects”. It was definitively correct for decades, however, along with unprecedented development of the STEM methodologies undertaken recently, true atomic resolution became possible, opening for more reliable quantification of crystallinity based entirely on STEM. Nevertheless, in our work we used a combination of several experimental techniques to monitor the structural quality (RBS/C, STEM, XRD, EBSD) providing data in excellent agreement with simulations. As such, as mentioned above, even though collecting the RBS/C data in γ -phase along different channels is spectacular, it is unlikely to change/add to the prime message of the paper. We wish the reviewer to consider the arguments above and hopefully waiving the request to add the RBS/C collected at the low index channels to this paper (for which we need to wait for additional experimental work to be done, unlikely changing the message of the paper, as such leading to unnecessary delay of the publication).

Comment: 3): *Related with the previous point, it is important to make clear that “radiation resistance” in the present paper is used in the sense of high threshold for amorphisation. Probably the authors could comment on displacement energies for different phases based on MD simulations. Also, it would be interesting to review literature on some ceramics (nuclear materials), where high*

amorphisation thresholds are found for spinels and irradiation of pyrochlores can lead to amorphisation or phase transition depending on the cation and anion radii.

Reply: We agree with the reviewer. Indeed, we use the term “radiation tolerance” to describe the ability of the materials to withstand high ion fluences generating a big number of defects, as we clearly state it throughout the paper, see also the arguments answering the first comment of this reviewer. Further, the aspect related to the threshold energies is overlapping with the first comment from reviewer #2. In reality, there are no experimental values of the displacement energies for Ga₂O₃ polymorphs. Even for the most intensively studied β -phase, we are aware of only one recent theoretical paper on the displacement energies (*J. Appl. Phys.* 133, 015703 (2023)). In particular, using MD simulations the authors determined the displacement energies to be 28 and 14 eV for Ga and O sublattices, respectively. In our calculation we used the default values of the displacement energies from the SRIM code and, specifically, for O the value is a twice higher as compared to that obtained in *J. Appl. Phys.* 133, 015703 (2023). According to the calculations, the lower values of the displacement energies results in the higher concentration of the primary defects by a factor of 1.55 as compared to those done with SRIM default values of displacement energies (see Fig. R1 below). Thus, the data obtained with the SRIM default values can be considered as a lower limit of the dpa thresholds. In addition, we would like to point out that there are no theoretical/experimental estimations of the displacement energies in γ -phase. Thus, at present stage, the dpa calculations used in this paper, might be considered as a way to normalize the irradiation conditions and to be able to compare the results obtained for different ions, rather than exact values. In the revised ms we have added a brief discussion in Suppl. Inf. on p. 3 and new ref S4.

Fig. R1 Comparison of the depth profiles for the total vacancies per implanted ion in beta-phase, as calculated using SRIM code for 400 keV Ni ions using two different sets of the displacement energies as indicated in the legend.

Regarding the idea of including a perspective/comparison with other classes of materials exhibiting high degree of resistance to amorphization. We think it is an interesting idea, however, to undertake a systematic comparison with different sorts of materials will require significantly more space and somewhat defocus the paper. In fact, we think the comparison between several groups of semiconductor materials in Fig. 1(d) is made exactly to fulfill this generalization function.

Comment: 4): *To the best of my knowledge, the first mentioning of a possible phase transition of (010) beta Ga₂O₃ upon ion implantation was published by Wendler et al. (NIM B 379 (2016) 85–90) and should be cited here. On the other hand, Lorenz et al. (Proc. SPIE 8987 (2014) 89870M) reported quite distinct RBS/C pattern in (100) oriented beta Ga₂O₃, where the aligned spectra reach the random level suggesting amorphisation starting at the surface. The authors themselves have published previous papers showing distinct strain accumulation upon ion implantation in Ga₂O₃ with different surface orientation. How can these results be related with the mechanism reported in the present paper?*

Reply: We are aware of these pioneering papers on ion implantation of β -Ga₂O₃. We included these refs. in the Suppl. Inf. of the revised version of the manuscript (p. 5 and new refs. S5 and S6). Notably, Wendler *et al.* indeed suggested a hypothesis that the observed disorder behavior can be related to phase transitions; however, without specific characterization/interpretation attempts. The present data – in this paper – confirm that the β -phase does not amorphize by irradiation, transforming instead to the γ -phase; further, this newly formed γ -phase is non-amorphizable, if not chemical effects are involved. In in ref. 14 we studied the process of the phase transformation alone and were not aware of the unprecedented radiation tolerance of the γ -phase – which is the main message of the present paper. As such, the hypothesis initially suggested by Wendler *et al.*, as well as mechanisms described in ref.14 and in the present paper are all valid and interrelated.

Comment: 5): *It would help to more easily understand the paper if the crystallographic relationship between beta and gamma phases would be mentioned explicitly early in the manuscript (e.g. on page 6 when discussing SAED patterns). Also, the orientation of the original crystal should be mentioned earlier (at this moment it is only mentioned in the methods section).*

Reply: We agree and included the orientation of the original Ga₂O₃ wafer in the caption of Fig. 1. Note that we mentioned the relationship between β/γ phases on p. 4 of the original manuscript too.

Comment: 6): *I find Fig. 3c quite confusing to read and you are losing the information on pair distance in this representation. In fact, from this figure it seems that similarities are only strong in the end of the 2nd shell window. Would it be clearer to plot the ratio PRDF_beta/PRDF_gamma as a function of the pair distance? It should be close to 1 after phase transformation. Similar figures are also shown in the supplement.*

Reply: The choice of the visualization option for the scattered diagonal-line plot in Fig. 3c is indeed a good question. Indeed, the unconnected scattered points may not effectively convey the information of the pair distance. The reason to choose such representation was because it can clearly show the correlation between the curves in comparison, as we further emphasized the Pearson correlation coefficient, Pr in Fig. 3d. As an alternative, we may use a modified representation plotting the ratio of the PRDF(β) to PRDF(γ) in a linear-log plot in Fig. R2.

As anticipated by the reviewer, the curves are close to 1 after the phase transition. However, because the PRDF(γ) exhibits nearly zero value in the beginning of the 2nd shell window, even minor changes in the PRDF(β) can be significantly amplified in the ratio plot. Consequently, this representation could potentially lead to a confusing interpretation that “ $\beta + 100$ FPs” curve exhibits overall higher similarity to the PRDF(γ), which is obviously not the case. In fact, just as accurately noted by the reviewer, the β -to- γ phase transition is seen in the β -Ga PRDF curves as the two

vanishing shallow peaks at the 4.5 and 4.75 Å, and an emerging peak around 5.0~5.2 Å (in the end of the 2nd shell window). Thus, taking all this aspects into consideration, we believe that the scattered diagonal-line plot can serve best in delivering our key message to the readers.

Fig. R2. Modified representation of the data in Fig. 3c in a linear-log ratio plot.

Comment: 7): *The two texts (main manuscript and supplement) would be easier to read with less cross references (which force the reader to read both documents in parallel). E.g. in Fig. 1S one could add the implantation energies in the figure labels (or caption) or in Fig. S7 one could repeat the color code in the caption instead of referring to the main manuscript.*

Reply: Figs. S1 and S7 were modified in accordance with the reviewer suggestions.

Comment: 8): *Supplement page 10: “The fractional numbers indicate that there is an artefact related to the misalignment...” Not clear what the authors want to say by “artefact”.*

Reply: For clarity and in order to avoid misunderstanding we suggest to replace the statement criticized by the reviewer by “**The fractional numbers indicate that there is misalignment between some low indexed directions of the β -Ga₂O₃ and the γ -Ga₂O₃ polymorphs**”. That means, for example, that the angle between β -Ga₂O₃ [0 2 -1] and β -Ga₂O₃ [0 2 1] is not exactly 90°, while the angle between γ -Ga₂O₃ [100] and γ -Ga₂O₃ [001] is exactly 90°. Thus, the directions of γ -Ga₂O₃ exhibit a mismatch with respect to β -Ga₂O₃ [0 2 -1] and β -Ga₂O₃ [0 2 1].

Comment: 9): *Supplement page 10: “(i) interface stacking in γ/β -Ga₂O₃ double polymorph structures is independent on the ion type used in fabrication process” Not clear from which technique you draw this conclusion since EBSD patterns are only shown for one sample.*

Reply: This conclusion is based not on the EBSD results alone and even not primarily; it is supported by systematic STEM analysis in the first place.

Comment: 10): Supplement page 11: “Nevertheless, there should be a reason for the DPA threshold overestimation deeper in the bulk in Fig. S5.” Not clear what you want to say here. I don’t think you want to say that SRIM overestimates DPA in depth.

Reply: We agree that the way how it was formulated may lead to misunderstanding similar to that mentioned by the reviewer in the example. In the revised manuscript we reformulated this sentence as following: “Nevertheless, there should be a reason for the distinct increase of the dpa threshold with an increase of the fluence and, therefore, advancing the interface deeper in the bulk in Fig. S5”.

Comment: 11): It seems that Fig. S11 is not mentioned in the text.

Reply: We note that firstly, the Fig. S11 is briefly mentioned in the original main text when discussing Fig. 3b (on page 7): “Additionally Fig. 3(b) illustrates the structural differences in β -Ga₂O₃ (up) and γ -Ga₂O₃ (down) before and after introduction of 600 FPs, where the dramatic changes— compared to the initial cell—are seen only in β -Ga₂O₃ (SI-VIII, Fig. S11).” Indeed, this text can be easily overlooked, therefore, we add one more sentence for the brief introduction of the Fig. S11. “... only in β -Ga₂O₃ (for the more detailed transition process, see SI-VIII, Fig. S11).”

In addition, we now add one paragraph in the supplementary materials to discuss the Fig. S11: “Fig. S11 illustrates the detailed snapshots for every 100-FPs step both in β and γ phases. As seen from Fig.S11, the O-sublattice retains the fcc stacking perfectly. The β -Ga sublattice gradually transfers to γ -like configuration, whereas no significant change is detected in the γ -Ga sublattice.”

Comment: 12): Supplement Fig. S13: “to exclude coloring of atoms around single vacancies, which do not distort the lattice” Maybe I do not understand what authors want to say here, but there can be a strong relaxation of the lattice around a vacancy.

Reply: Thanks for pointing out this potentially misleading text. We address this issue together with the comment #1 from the reviewer #3. The caption of Fig. S13 was modified for clarity.

Comment: 13): Some typos should be corrected: e.g. interface instead of interphase; fluences instead of fluencies; page S17: caused by the cumulated (instead of causing); page S18 Fig. S13 shows the MSD (not Fig. S12); page S19: in the same way as that in Fig (instead of than of that); page 10 Coulombic (instead of Columbic); page 15 density of particles in the system

Reply: Thank you. In the revised manuscript we have fixed all the typos.

Response on the comments of the Reviewer #2:

Comment: 1) they use values for the displacement energies of Ga and O that are relatively close to each other, but a recent paper using molecular dynamics simulations, based on first principles density-functional methods, to determine the nature and stability of the defects generated by atoms knocked-out by particle irradiation at near threshold energies (found to be $28\pm 1\text{eV}$ for Ga and $14\pm 1\text{eV}$ for O). Blair R. Tuttle, Nathaniel J. Karom, Andrew O’Hara, Ronald D. Schrimpf, and

Sokrates T. Pantelides, "Atomic-displacement threshold energies and defect generation in irradiated β -Ga₂O₃: A first-principles investigation", *Journal of Applied Physics* 133, 015703 (2023). <https://doi.org/10.1063/5.0124285>

Reply: see the reply to the comment 3 of reviewer #1.

Comment: 2) *the damage accumulation in WBG and UWBG semiconductors is known to be very temperature-sensitive. You need to discuss more details on how the temperature of the samples was controlled during the implantation steps. Given the low thermal conductivity of the beta Ga₂O₃, if you are seeing significant temperature excursions, it could imply the gamma polytype is above its threshold temperature where amorphization cannot occur, as also happens with GaN and AlN.*

Reply: Despite we have not controlled the sample temperature during the implantation in this work, we used relatively low beam currents for all implants (not exceeding 1 $\mu\text{A}/\text{cm}^2$ for Ni/Ne and 0.1 $\mu\text{A}/\text{cm}^2$ for Ga/Au ions). Thus, the risk of significant increase of the sample temperature during implantation was low (we added this in Methods section of the revised ms). Furthermore, our previous results (*Appl. Phys. Lett.* 118, 232101 (2021)) indicate that the temperature in excess of 100 °C are required to have a noticeable effect on the disorder formation in β -Ga₂O₃ samples as compared to the room temperature implants.

Comment: 3) *a comment might be added about the Ni precipitates formed. Just in general, the very high doses used are above any of those typical for device fabrication, which means that the high radiation resistance gamma region would not have much utility, since it is both non-stoichiometric and thermally unstable.*

Reply: We agree with the reviewer that introducing self or foreign atoms will end up in stoichiometry disturbances and/or precipitation formation in the material for high enough ion doses. However, we should point out that the formation of γ -phase is observed for relatively high (1e16 Ni/cm²) dose but this ion dose is not high enough for the formation of metallic precipitates. Furthermore, we demonstrated that phase transformation is related to the disorder threshold, and not to the high concentration of the implanted ions. This implies that implantation of heavier atoms, leads to the formation of γ -phase at lower doses. As it comes to the statement that the “radiation resistance gamma region would not have much utility, since it is both non-stoichiometric and thermally unstable” it remains to be investigated. Perhaps, the γ -phase fabricated by Ga implants resulting into Ga-excess crystalline gamma/beta structures with presumably low Ga-vacancy content (due to Ga-excess) may be an interesting candidate for more detailed device studies.

Comment: 4) *SRIM has limitations in that it doesn't treat defect recombination during dynamic annealing. Please add a brief comment about the expected discrepancy in dpa from SRIM versus what might actually be the accumulated disorder level. Can you use your RBS spectra to make this clearer?*

Reply: Indeed, we use SRIM to calculate the number of primary defects which is expressed in dpa, while the actual disorder level we estimate in relative units from the RBS/C data. Potentially, we can estimate the dynamic defect annealing efficiency for the low implanted doses where phase transitions have yet been ignited. However, these estimations will be very speculative since the defect accumulation in β -Ga₂O₃ is very sensitive to the balance between primary defect

generation/recombination rates determined by the ion flux, irradiation temperature and collision cascade density effects (*APL 118 (2021), 232101*). Thus, instead of doing speculative estimates we propose to keep this part of the discussion the way it is.

Comment: 5) *What is the role if any in electronic excitation versus nuclear displacements, since most of the energy deposited is from the former.*

Reply: That is a very interesting question since the defect formation under these two processes can be different indeed. So far, in all our experiments we used the energy range where the nuclear energy deposition dominates. Moreover, we did not consider any electronic excitations in the performed simulations. Obviously, swift heavy ion irradiation tests are required to reveal the impact of the electronic excitation on the disorder formation/phase transitions in Ga₂O₃. We are planning systematic experiments on that matter, but consider it as its own independent work, as such out of scope for the present paper.

Response on the comments of the Reviewer #3:

Comment: 1) *One technical comment related to the sentence on page 8: "In these simulations, we see the O sublattice is highly rigid and strongly prone to recrystallization into face-centered-cubic (fcc) stacking, despite higher mobility of O atoms compared to Ga during the active phase of cascades". Is it "despite the higher mobility of O atoms" or is it because of the higher mobility of the O atoms? The latter may be the more correct explanation. This is a little confusing, and the authors should consider rewording to provide clarify.*

Reply: Indeed, it may sound confusing to a reader, although we specifically emphasized that the high mobility of O ions is only observed during the active phase of high-energy displacements, far from thermal equilibrium conditions. In this phase, the mass of the atom contributes stronger to the magnitude of the displacement than the ionic radius, which plays more active role during the relaxation period. To clarify, we propose a modification: “**In these simulations, we see that the O sublattice is highly rigid and strongly prone to recrystallization into face-centered-cubic (fcc) stacking, despite stronger displacement of O atoms compared to Ga during the active periods of cascades, see Fig. S13**”. Correspondingly, we modified the caption of Fig. S13. In response to the reviewer #1 comment on coloring of O atoms around a single vacancy, we rephrase in the caption of Fig. S13b too.

Comment: 2) *A minor comment is that "displacements per atom" is a unit of damage dose and generally written in lower case "dpa" not upper case "DPA". In Figure S1(a), the equation for DPA should be written as a function of Fluence, not Dose. Likewise, in the inset in Figure S1(b), the dpa should be a function of Fluence (cm⁻²), not Dose. This should also be corrected in the associated Supplementary Material text. The authors use the term "fluence" elsewhere and may confuse readers by substituting the term "dose" for "fluence". They are not the same.*

Reply: Thank you. In order to avoid any misunderstanding we use only the term “fluence” in the revised version of the paper. In addition, we changed to the lower case “dpa” throughout the paper.

REVIEWERS' COMMENTS

Reviewer #1 (Remarks to the Author):

I apologize to the authors for my word "pompous", I used it in the sense of unnecessarily glamorous. Reading a title "Universal radiation tolerant semiconductor" I expect a semiconductor which is radiation tolerant and universally usable for many applications (a better silicon). I understand the arguments of the authors, so if the editor is fine with the title I do not insist in changing it.

Otherwise the authors have responded to my comments to the best of their possibilities. So, I recommend the paper for publishing.

Reviewer #2 (Remarks to the Author):

I'm happy with the revisions made by the authors. It's not often there is a completely unexpected result in semiconductor materials and the extraordinary resistance of the gamma polymorph of Ga₂O₃ reported here is noteworthy. Whether or not there is utility for this polymorph will come when experiments to determine its dopability are carried out, ie. whether the electrical properties can be manipulated to match the ion irradiation stability.

It's a result that will attract attention from beyond the wide bandgap semiconductor community. There is a typo in the Supplemental file, on the second page, ie. "there are no any experimental values of the displacement energies in Ga₂O₃ polymorphs".

Reviewer #3 (Remarks to the Author):

The authors have adequately addressed all my comments in the revised manuscript.